# ssGBLUP Method Improves the Accuracy of Breeding Value Prediction in Huacaya Alpaca

**DOI:** 10.3390/ani11113052

**Published:** 2021-10-26

**Authors:** Betsy Mancisidor, Alan Cruz, Gustavo Gutiérrez, Alonso Burgos, Jonathan Alejandro Morón, Maria Wurzinger, Juan Pablo Gutiérrez

**Affiliations:** 1Departamento de Producción Animal, Universidad Nacional Agraria La Molina, Lima 12056, Peru; 20140358@lamolina.edu.pe (B.M.); gustavogr@lamolina.edu.pe (G.G.); jmoron@lamolina.edu.pe (J.A.M.); mwurzinger@lamolina.edu.pe (M.W.); 2Centro Genético de Pacomarca–Inca Tops S.A., Miguel Forga 348, Arequipa 04001, Peru; aburgos@pacomarca.com; 3Departamento de Producción Animal, Universidad Complutense de Madrid, E-28040 Madrid, Spain; gutgar@vet.ucm.es

**Keywords:** alpaca, genomic selection, SNP markers

## Abstract

**Simple Summary:**

Alpaca breeding takes place in the most entrenched areas of the Andes, where the conditions to implement genetic improvement programs are very difficult. Likewise, taking phenotypic records is limited in its ability to predict genetic merit accurately. For this reason, genomic information is shown as an alternative that helps to predict the genetic values of fiber traits more precisely. This study showed how genomic information increased precision by 2.623% for the fiber diameter, 6.442% for the standard deviation of the fiber diameter, and 1.471% for the percentage of medullation compared to traditional methods for predicting genetic merit, suggesting that adding genomic data in prediction models could be beneficial for alpaca breeding programs in the future.

**Abstract:**

Improving textile characteristics is the main objective of alpaca breeding. A recently developed SNP chip for alpacas could potentially be used to implement genomic selection and accelerate genetic progress. Therefore, this study aimed to compare the increase in prediction accuracy of three important fiber traits: fiber diameter (FD), standard deviation of fiber diameter (SD), and percentage of medullation (PM) in Huacaya alpacas. The data contains a total pedigree of 12,431 animals, 24,169 records for FD and SD, and 8386 records for PM and 60,624 SNP markers for each of the 431 genotyped animals of the Pacomarca Genetic Center. Prediction accuracy of breeding values was compared between a classical BLUP and a single-step Genomic BLUP (ssGBLUP). Deregressed phenotypes were predicted. The accuracies of the genetic and genomic values were calculated using the correlation between the predicted breeding values and the deregressed values of 100 randomly selected animals from the genotyped ones. Fifty replicates were carried out. Accuracies with ssGBLUP improved by 2.623%, 6.442%, and 1.471% on average for FD, SD, and PM, respectively, compared to the BLUP method. The increase in accuracy was relevant, suggesting that adding genomic data could benefit alpaca breeding programs.

## 1. Introduction

The purpose of alpaca breeding is to improve the textile properties of the fiber [1]. The textile industry seeks quality fibers [2], measured by the fineness and the low variability of its diameter [3]. Furthermore, it is known that reducing or eliminating the so-called itching factor would enhance its economic value, which could be achieved by decreasing the percentage of medullation [2]. Therefore, improving the fiber’s textile characteristics consists of producing a finer fiber with low variability and a lower itching factor. 

Currently, in alpaca breeding, most advanced improvement programs rely on quantitative genetics using Best Linear Unbiased Prediction (BLUP) methods with a multi-trait animal model with repeated measurements [2]. Fiber traits such as fiber diameter (FD), standard deviation of fiber diameter (SD), and percentage of medullation (PM) can be measured multiple times in the animal, usually at shearing time. However, it has been shown that non-genetic factors highly influence fiber characteristics. Fiber diameter tends to increase with the animal’s age [3] but is also affected by the stage of pregnancy and lactation [4]. 

Genomic selection implementation has recently evolved rapidly in many species. Meuwissen et al. [5] proposed combining statistical models with genomic data to help increase the accuracy of breeding value estimates. Legarra et al. [6] and Christensen and Lund [7] reported how to integrate the information of the additive relationship matrix (**A**) and the genomic relationships matrix (**G**) in a combined matrix (**H**) and to develop a single-step Genomic BLUP (ssGBLUP) method. Matrix **H** can be understood as a modification of the regular pedigree relationships by including genomic relationships [8]. The ssGBLUP integrates all the phenotypic, pedigree, and genomic information available simultaneously to predict genomic merit values for genotyped and non-genotyped individuals through the combined matrix **H** [9,10,11]. This allows the use of all the available information in a genetic improvement program; many species rely on this new way of predicting breeding values [11,12,13]. Possible advantages of genomic information are to increase the accuracy of genetic merit, reduce the generational interval, and evaluate traits that are difficult to measure [11,14,15]. Hence, alpaca breeding programs could access these advantages with the use of genomic selection. However, this has not been possible because there was no specific genomic beadchip for alpacas until recently. This genomic beadchip has just been built and with it, it has been possible to establish a map of 76,508 Single Nucleotide Polymorphism (SNP) markers [16]. Therefore, the objective of this work was to compare, for the first time, the prediction accuracy of three important fiber traits: fiber diameter (FD), standard deviation of fiber diameter (SD), and percentage of medullation (PM) using BLUP and ssGBLUP methods in Huacaya alpacas. 

## 2. Materials and Methods

The data were obtained from the PacoPro v5.10 software from the Pacomarca Genetic Center, which contains pedigree information from 1992 to 2020 and phenotypic data collected from 2001 to 2019. The fiber traits were the FD and SD as described by Gutiérrez et al. [1], and PM as described by Cruz et al. [2]. 

Table 1 summarizes the information available for the analysis. The number of records was 24,169 for FD and SD and 8386 for PM from a total pedigree of 12,431 animals, of which 6889 had their own performance records. The number of genotyped animals was 431, which had 2774 records for FD and SD, and 1767 for PM. The number of animals with offspring was 1943, and the number of animals with offspring and phenotypic information was 1867. The number of animals with unknown sires was 1554, the number of animals with unknown dams was 1278, and the number of animals with both unknown parents was 1254. The total number of sires was 246, the number of sires with progeny in the data was 246, and the number of sires with records and progeny in the data was 212. The number of dams was 1697, the number of dams with progeny in the data was 1692, and the number of dams with their own records and progeny in the data was 1655. The average of the records for FD was 22.82 µm, 5.38 µm for SD, and 47.75% for PM.

A recently developed DNA microarray consisting of 76,508 SNPs from 37 pairs of chromosomes, 36 autosomal chromosomes, and one sex chromosome [16], was used to genotype 431 alpacas. For quality control, all SNPs with a call rate ≤ 95% and minor allelic frequency (MAF) ≤ 0.05 were removed, leaving 60,624 SNP markers.

Genetic parameters were, firstly, estimated for the three traits, and then the precision of the breeding values was analyzed comparing two methodologies. Two methods were used: A traditional BLUP method with phenotypic data and pedigree-based relationship matrix (**A**) and an ssGBLUP method based on a combined matrix (**H**) constructed from a matrix **A** and a genomic relationship matrix (**G**). Variance components were estimated using restricted maximum likelihood (REML) [17,18].

FD, SD, and PM traits were independently analyzed. For each of them, deregressed phenotypes were called **ď**. They were assumed to be the true breeding values for prediction, averaging the regressed values of an individual when there were repeated measurements of the phenotype. These were obtained by fitting the fixed effect solutions estimated with the equation y=Xb+e in which contemporary groups defined by year of sampling (19 levels in FD and SD, 4 levels in PM), color (9 levels) [19], age in days as a linear and quadratic covariate, and sex by physiological status were defined as the fixed effects in **b,** and **X** was the corresponding incidence matrix. The sex by physiological effect was defined as one level for males and two for females (empty or lactating). The vector **y** was FD, SD, or PM phenotypes.

BLUP and ssGBLUP performance were compared by cross-validation as predictors of the de-regressed phenotypes **ď** using the model equation y=Xb+Zu+Wp+e, with (co)variance matrices:$$\left(\begin{array}{c} u\\ c\\ e \end{array}\right)\sim N\left(\begin{array}{c} 0\\ 0\\ 0 \end{array},\left[\begin{array}{ccc} M\sigma_{u}^{2} & 0 & 0\\ 0 & I_{c}\sigma_{c}^{2} & 0\\ 0 & 0 & I_{e}\sigma_{e}^{2} \end{array}\right]\right)$$

**M** was equal to $A\sigma_{u}^{2}$ in BLUP and $H\sigma_{u}^{2}$ in ssGBLUP, $\sigma_{u}^{2}$ is the additive genetic variance, $\sigma_{c}^{2}$ is the permanent environmental variance, $\sigma_{e}^{2}$ is the residual variance, and the heritability (h^2^) is $h^{2}=\frac{\sigma_{u}^{2}}{\left(\sigma_{u}^{2}+\sigma_{c}^{2}+\sigma_{e}^{2}\right)}$. **I_c_** is the identity matrix of order equal to the number of permanent environmental subclasses, **I_e_** is the identity matrix of order equal to the number of records, **A** is the numerator relationship matrix according to the pedigree information, and **H** is a similar matrix to **A** that includes both pedigree-based relationships and differences between pedigree-based and genomic-based relationships [20]. The ssGBLUP method is a modification of BLUP in which the numerator relationship matrix **A**^−1^ matrix must be replaced by **H**^−1^ [12], such as:
$$H^{-1}=A^{-1}+\left[\begin{array}{cc} 0 & 0\\ 0 & G^{-1}-A_{22}^{-1} \end{array}\right]$$
where **A**_22_ is a numerator relationship matrix for genotyped animals, and **G** is a genomic relationship matrix [14]. **G**^−1^ was obtained as the inverse of a combination of the **G** matrix and the corresponding **A** matrix, weighting them by 0.95 and 0.05, respectively [14].

Records from a random sample of 100 animals out of the 431 genotyped ones were initially removed to be used later as the testing set, keeping the rest as a training set. BLUP and ssGBLUP were compared by computing the correlation between the true breeding values **ď** defined above and the predicted breeding values obtained both by BLUP (PBV) and by ssGBLUP (GPBV). This procedure was carried out in fifty replicates.

An additional analysis to check the use of genomic selection in non-genotyped animals was carried out. Phenotypes of 390 animals of the last year of recording were initially removed. Later the corresponding deregresed phenotypes were predicted from both BLUP and ssGBLUP methodologies and their accuracies were compared.

In the next step, PBVs and GPVBs of genotyped animals were compared using a Pearson correlation. Given that selected animals can vary when selected by truncation using different criteria, the ranking of all animals using the two methods was compared by calculating a Spearman’s rank correlation between them. To better illustrate this, selection by truncation was simulated both with the PBV and GPBV, where genotyped animals were ranked by their genetic value for each trait, and the top 25% were selected (108 animals). Thus, the percentage of the same animals that would be selected in both methods was calculated for each trait.

The SNP markers’ quality control was carried out using the R language [21]. The *lm*, *cor*, *cor.test* and *t.test* functions of the R language [21] were used. The RENUMF90, REMLF90, and BLUPF90 programs [22] were used for the variance component estimation and the genetic evaluations.

## 3. Results

The mean heritabilities estimated across replicates by BLUP and ssGBLUP methods were not significantly different for FD, SD, and PM traits, as shown in Table 2. The heritabilities for FD were moderate at 0.334 ± 0.001 for BLUP and 0.336 ± 0.001 for ssGBLUP. The heritabilities for SD were moderate at 0.381 ± 0.001 for BLUP and 0.382 ± 0.001 for ssGBLUP, and moderate to low for PM at 0.158 ± 0.001 for BLUP and 0.160 ± 0.001 for ssGBLUP. 

The mean prediction accuracy for FD, SD, and PM traits under BLUP and ssGBLUP methodologies is shown in Table 3. The BLUP mean (±standard error) accuracies were 0.505 (±0.015), 0.445 (±0.019) and 0.308 (±0.017) for FD, SD, and PM, respectively. The ssGBLUP mean (±standard error) accuracies were 0.517 (±0.011), 0.472 (±0.015) and 0.311 (±0.013) for FD, SD, and PM, respectively. The increases in accuracy from ssGBLUP compared to BLUP expressed as a percentage of the BLUP accuracies were 2.623%, 6.442%, and 1.471% for FD, SD, and PM, respectively, non-significant only for the PM trait (*p* > 0.05).

The Pearson correlations of genetic values for fiber traits between BLUP and ssGBLUP were 0.975 for FD, 0.976 for SD, and 0.959 for PM. Meanwhile, the Spearman’s correlations show how much the individuals conserve their position in a ranking. They were 0.972 for FD, 0.970 for SD, and 0.956 for PM.

The percentage of animals for FD, SD, and PM that would be selected using both methods with truncation selection of the best 25% of the animals is shown in Figure 1. The number of animals selected using both methods is 93/108 for FD, 95/108 for SD, and 90/108 for PM.

Analyses comparing the accuracies of BLUP to ssGBLUP in non-genotyped animals showed that predictive ability increased by 1.331% and 0.522% for FD and SD, respectively, but decreased 46.288% for the PM trait.

## 4. Discussion

Alpaca management systems in Peru are traditional and are based on the exploitation of the Andean rangelands, where the management of other domestic animal species is difficult [23]. In most Peruvian production systems for different livestock species, breeding programs are not established. This practice does not represent a big problem in other species, where genetics can be imported. However, Peru has 90% of the worldwide alpaca population, which makes alpaca breeding an important tool for Peru and global animal farming.

Genomics is undoubtedly a tool of increasing interest that has different uses in animal management [24]. Genomic selection is relevant when considering a breeding program scheme, as other domestic species do [15], and alpaca breeding can be complemented with this new methodology. Therefore, this first study, which addressed the utility of using SNP markers, can help encourage the adoption of this new way of selection when considering an improvement program. The deficiency in the data collection of both phenotype and pedigree, or a much earlier evaluation when the animals do not yet have phenotypic data, can be perfomed with much greater accuracy if we rely on molecular markers.

The first studies that evidenced the importance of using molecular genetics regarding fiber traits were reported by Pérez-Cabal et al. [25]. However, the first approximation of the use of markers was reported by Mamani et al. [26], where it was possible to cover 40% of the alpaca genome. Subsequently, the identification and location of some genes related to the genetic control of the diameter and color of the coat were described by Mendoza et al. [27].

This study showed that heritabilities estimated by BLUP and ssGBLUP were similar to those reported by Cruz et al. [2], in which they analyzed the FD, SD, and PM traits, but jointly with other production traits. On the other hand, there did not seem to be a great difference between the heritabilities estimated by BLUP and ssGBLUP methods, as the differences were not statistically significant (Table 2). Variations have been reported that may be influenced by the repeatability of the data, where sometimes the heritability estimated by the BLUP method may be greater, less, or equal compared to the ssGBLUP method [10,28,29].

The ssGBLUP methodology provided greater accuracy of prediction than BLUP, as reported in the case of carcass traits in chicken broilers [10,30,31]. The use of genomic information (ssGBLUP) generates significantly greater accuracy than the traditional BLUP for FD and SD traits since genomic relationships are more accurate than relationships based on pedigree [32]. The marker panel can detect small genetic variations since the ssGBLUP methodology can integrate the phenotypic, genomic, and pedigree information [10].

However, it must be considered that these comparisons are not directly comparable to our results due to the differences between the size of the genotyped population, pedigree size, and the number of records in other species. In carcass traits of sheep, the increase in accuracy was 33.3% [33]. In dairy sheep for total milk yield, the accuracy increased to 47.98% [34], while in dairy goats, the increase in accuracy was from 5% to 7% for milk traits [35] In chicken broilers, the increase was around 22% in growth-related and carcass traits [10]. Accuracy of genomic evaluation depends on several factors, including linkage disequilibrium (LD) between markers and quantitative trait loci (QTL), effective population size (Ne), in addition to the relationship between individuals in the training and validation data [36,37,38]. Additionally, the reference population size [33,39] and its composition [40], the size of the training data [41], and heritability [38,41] are factors to be considered.

In the present study, the increase in accuracy was relevant even when the number of genotyped animals was small (431 animals). The reference population size affects the prediction accuracy of breeding values [40,42]. Accuracy tends to increase even more if the size of the reference population continues to grow over the years [33,35], which suggests that, in alpacas, a more significant increase in accuracy should be possible if the reference population is increased.

The results comparing the predictive ability on non-genotyped animals showed much less improvement in the new methodology for FD and SD traits. This highlights the importance of genotyping the selection candidates. In fact, the prediction ability for the PM trait, with low heritability and fewer records, considerably worsened.

Song et al. [11] explained that in real data, around 400 genotyped reference animals probably could not provide more additional information compared to the pedigree information consisting of 5000 individuals. In the present study, there were 431 genotyped animals against 12,431 animals with pedigree information (Table 1); however, the increase in precision was noticeable at 2.623%, 6.442%, and 1.471% for FD, SD, and PM, respectively. This result can be explained by the fact that in the present study, animals that had second shearing data were the ones selected for genotyping, since in alpacas, it has been shown that the second fiber evaluation, approximately when the alpaca is 2 years old, will be related to the true genetic potential of each animal [6]. Lourenco et al. [40] highlighted the importance of having a good reference population with consistent phenotypic information when predicting the genomic values of young animals.

However, it should be noted that alpaca farming has not yet implemented artificial insemination as an intensive breeding technique, and natural mating is a common practice. This practice leads to the need for a high number of males to cover reproductive needs. Therefore, the ranking variation may be less influential due to the low number of females assigned to each male in the breeding campaign [43]. Our results show only an overlap in selected animals of 86.11%, 87.96%, 83.33% for FD, SD, and PM, respectively, using PBV and GPBV (Figure 1). However, the use of the genomic information can help to increase the accuracy of the breeding values.

Genomic selection works best in populations where the effect of the marker comes from animals with data and whose marker-selected progeny go on to produce phenotype data and reenter the training population, which then becomes dynamic. Additionally, considering that the cost of genotyping in alpacas is high and that prediction accuracy is much higher when marker and phenotype information complement each other, data collection is still an important tool when implementing genomic selection. The eventual cost reduction may make its implementation more feasible.

## 5. Conclusions

The use of genomic selection increased the accuracy of the genetic value estimation through the ssGBLUP methodology, showing its potential for the genetic improvement of fiber traits. Therefore, ssGBLUP is a new methodology to predict genetic values, especially for difficult-to-measure traits, such as the percentage of medullation. Moreover, it is necessary to genotype more animals to know the real possibilities of genomic selection in alpacas. Finally, it is important to encourage the recording of information from a larger population to extend the benefits of this methodology to a larger population and, thus, have more impact on alpacas.

## Figures and Tables

**Figure 1 animals-11-03052-f001:**
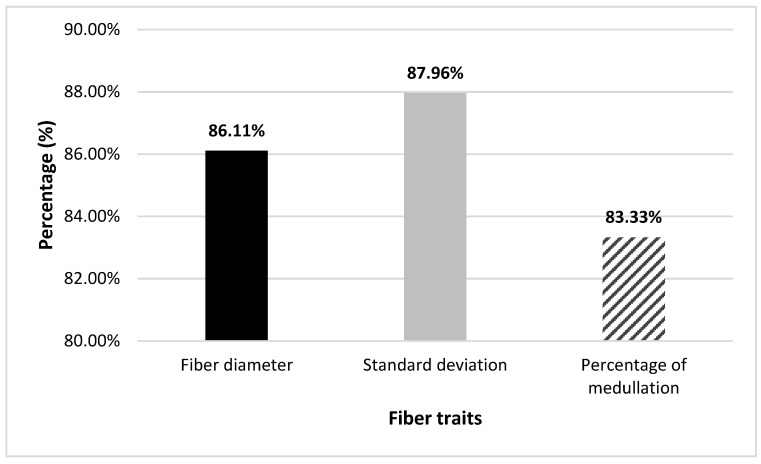
Percentage of same selected animals for fiber diameter (FD), standard deviation (SD) and percentage of medullation (PM) using BLUP and ssGBLUP methods by truncation selection of top 25%.

**Table 1 animals-11-03052-t001:** Number of records of the fiber diameter (FD), standard deviation (SD) and percentage of medullation (PM) and pedigree records in the database.

	Animals (n)	Total Records		
		FD	SD	PM
Full pedigree	12,431			
Animal with records	6889	24,169	24,169	8386
Genotyped animals	431	2774	2774	1767

**Table 2 animals-11-03052-t002:** Mean of variance components, heritabilities (h^2^) ± standard error of 50 replicates estimated by BLUP and ssGBLUP methods, and significance of the differences in h^2^ between methods for fiber diameter (FD), standard deviation (SD) and percentage of medullation (PM) in Huacaya alpacas.

Methodology	Traits	$\sigma_{u}^{2}$	$\sigma_{c}^{2}$	$\sigma_{e}^{2}$	h^2^
BLUP	FD	2.824 ^***^ ± 0.002	1.289 ^***^ ± 0.002	4.332 ^ns^ ± 0.001	0.334 ^ns^ ± 0.001
	SD	0.354 ^ns^ ± 0.001	0.144 ^ns^ ± 0.001	0.431 ^ns^ ± 0.001	0.381 ^ns^ ± 0.001
	PM	27.416 ^***^ ± 0.089	39.509 ^***^ ±0.079	106.316 ^ns^ ± 0.128	0.158 ^ns^ ± 0.001
ssGBLUP	FD	2.842 ^***^ ± 0.002	1.280 ^***^ ± 0.002	4.331 ^ns^ ± 0.001	0.336 ^ns^ ± 0.001
	SD	0.355 ^ns^ ± 0.001	0.143 ^ns^ ± 0.001	0.431 ^ns^ ± 0.001	0.382 ^ns^ ± 0.001
	PM	27.717 ^***^ ± 0.089	38.965 ^***^ ± 0.077	106.352 ^ns^ ± 0.128	0.160 ^ns^ ± 0.001

BLUP= Best Linear Unbiased Prediction methodology, ssGBLUP = single-step Genomic BLUP methodology, $\sigma_{u}^{2}$= additive genetic variance, $\sigma_{c}^{2}$
= permanent environmental variance, $\sigma_{e}^{2}$
= residual variance, (^ns^) = non-significant, (***) = *p* < 0.001.

**Table 3 animals-11-03052-t003:** Mean prediction accuracy ± standard error of 50 replicates of prediction accuracy of genetic values (BLUP) and genomic values (ssGBLUP), the difference in accuracy between the methods expressed as a percentage in Huacaya alpacas.

Traits	BLUP ^1^	SsGBLUP ^2^	Difference	Increase (%)
FD	0.505 ± 0.015	0.517 ± 0.011	0.012 ^*^ ± 0.003	2.623
SD	0.445 ± 0.019	0.472 ± 0.015	0.027 ^**^ ± 0.004	6.442
PM	0.308 ± 0.017	0.311 ± 0.013	0.004 ^ns^ ± 0.003	1.471

^1^ BLUP = Best Linear Unbiased Prediction methodology, ^2^ ssGBLUP = single-step Genomic BLUP methodology, FD = fiber diameter, SD = standard deviation, PM= percentage of medullation, (^ns^) = non-significant, (*) = *p* < 0.05, (**) = *p* < 0.01.

## Data Availability

Not Applicable.

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
