# Peer review of "ssGBLUP Method Improves the Accuracy of Breeding Value Prediction in Huacaya Alpaca"

_animals, 2021, doi:10.3390/ani11113052_

Round 1

Reviewer 1 Report

This paper describes a fairly straightforward study comparing three alpaca traits using standard breeding (BLUP) and genomic (ss-GBLUP) testing. It has been written well and the English is good. Once my queries have been addressed the paper could be published.

  1. The claim is that only moderate differences were seen in the mean heritability’s (Table 2) but there is no way of substantiating this claim when no p values have been given between BLUP and ss-GBLUP for the three traits (FIber diameter, Standard deviation and Percentage of medullation). Please supply p values or some other statistical analysis appropriate for these groups.
  2. The claim is that differences were seen in the mean prediction accuracy (Table 3) but there is no way of substantiating this claim when no p values have been given between BLUP and ss-GBLUP for the three traits (FIber diameter, Standard deviation and Percentage of medullation). Please supply p values or some other statistical analysis appropriate for these groups. It is also confusing that values obtained for BLUP and ss-GBLUP have not been given a unit and that the increase has been expressed as a percentage and not in the same units as the raw values. Please explain or change this.
  3. Table 4 and Figure 1 need more explanation. What is the purpose of the Pearson and Spearman correlations? It is not clear what values have been correlated. Can these analyses be compared to the data in the previous 3 tables? Have the same data points been evaluated in all 4 tables and the figure or are they different analyses? Are the analyses in Table 4 and Figure 4 the statistical analysis that was asked for in the previous two points?

Reviewer 2 Report

This manuscript systematically explored the predictive performance of pedigree-based BLUP and SSGBLUP which could utilize both pedigree and genomic information by using three important fiber traits: fiber diameter, the standard deviation of fiber diameter, and percentage of medullation in Huacaya alpacas, the results showed that SSGBLUP outperformed BLUP for all three traits, indicating the big potential of SSGBLUP in the application of predicting the genetic merit for Huacaya alpacas. The overall experimental design, data quality control, and the structure of the current manuscript have been thorough, however, there are still some issues that should be addressed before it can be published, please find the comments below:

(1) For the construction of the H inverse matrix at line 124, I am wondering that G-1 is the direct inverse of the original genomic relationship matrix (G)? Most of the papers used a weighted (w) mixed component of a scaled G matrix (eg. w = 0.95) and the A matrix (eg. w = 0.05).

(2) Please briefly define all parameters in the formula at line 117, for example, the phenotypic variance, residual variance, and so on, and please subsequently describe how the heritability was calculated.

(3) As all the three traits have repeated records, I suggest listing all the variance components (additive genetic term, permanent environment term, and the residual term) and their corresponding heritabilities in Table 2 for a more intuitive comparison of the change of all the variance components for two models, rather than merely exhibiting the heritabilities of the traits.

(4) The advantage of SSGBLUP can not only be embodied in the genotyped individuals but also mainly in the non-genotyped individuals in pedigree, as the relationship between non-genotyped individuals can be adjusted by the genotyped individuals in H matrix, the authors only compared the predictive performance of 100 randomly selected genotyped individuals, i suggest to do additional work and provide the results showing the difference of the predictive performance of two models for non-genotyped individuals.

(5)For the traits with repeated records, one individual has only one predicted value for either PBVs or GPBVs, but the de-regressed phenotypes of this individual may have more than one records, author should give an explanation of how the predictive accuracy was calculated.

Round 2

Reviewer 1 Report

Table 2: Significance values ("*"; "**"; "*") should also be written in the table. Overall the presentation of the table needs improvement; i.e. the values of each FD, SD and PM should be written on one line (2.824+/-0.002 etc).

Author Response

We have added the significance values to the table, and we have arranged the value in a single line.

Reviewer 2 Report

Thank you for revising the manuscript. I have no additional comments.

Author Response

without changes